# Nitrophenyl Thiourea-Modified Polyethylenimine Colorimetric Sensor for Sulfate, Fluorine, and Acetate

**DOI:** 10.3390/s24123751

**Published:** 2024-06-09

**Authors:** Kediye Kuerbanjiang, Kuerbanjiang Rouzi, Si-Yu Zhang

**Affiliations:** 1College of Chemistry, Xinjiang University, Urumqi 830017, China; kadirye@stu.xju.edu.cn; 2Chemical Engineering Department, McGill University, Montreal, QC H3A 0C5, Canada; 3School of Chemical Engineering and Technology, Xinjiang University, Urumqi 830017, China; zsy@stu.xju.edu.cn; 4Key Laboratory of Oil and Gas Fine Chemicals, Ministry of Education & Xinjiang Uyghur Autonomous Region, Urumqi 830017, China

**Keywords:** polyethyleneimine, thiourea, colorimetric detection, anion recognition

## Abstract

A thiourea-based colorimetric sensor incorporating polyethyleneimine (PEI) and chromophoric nitrophenyl groups was synthesized and utilized for detecting various anions. Structural characterization of the sensor was accomplished using FTIR and 1H-NMR spectroscopy. The sensor’s interactions and colorimetric recognition capabilities with different anions, including CI^−^, Br^−^, I^−^, F^−^, NO_3_^−^, PF_6_^−^, AcO^−^, H_2_PO_4_^−^, PO_4_^3−^, and SO_4_^2−^, were investigated via visual observation and UV/vis spectroscopy. Upon adding SO_4_^2−^, F^−^, and AcO^−^ anions, the sensor exhibited distinct color changes from colorless to yellow and yellowish, while other anions did not induce significant color alterations. UV/vis spectroscopic titration experiments conducted in a DMSO/H_2_O solution (9:1 volume ratio) demonstrated the sensor’s selectivity toward SO_4_^2−^, F^−^, and AcO^−^. The data revealed that the formation of the main compounds and anion complexes was mediated by hydrogen bonding, leading to signal changes in the nitrophenyl thiourea-modified PEI spectrum.

## 1. Introduction

Anion recognition technology has a wide range of applications in various fields, due to its selective recognition and sensing functions. Anion acceptors are vital in chemistry, medicine, life science, environmental science, and other fields [1]. The different sizes and shapes of anions are responsible for the slow development of anion-binding agents compared with cation-binding agents [2,3,4]. The ‘naked-eye’ detection of anions without using spectroscopic instrumentation is an important area in the design and fabrication of new anion receptors. Naked-eye or optically detectable receptors have been developed via the covalent attachment of chromogenic or fluorogenic groups [5]. The anionic recognition groups are usually hydrogen bond donors, such as urea, thiourea, amide, amine, guanidine, pyrrole, phenolic hydroxyl groups, etc. The signal reporter groups are photosensitive or electrochemically active materials, thus establishing different anionic-type sensing modes [6,7].

The (thio)urea motif, characterized by two parallel NH groups flanking a carbonyl in a planar topology, demonstrates remarkable efficacy in binding trigonal planar anions such as oxoanions or carboxylates [8,9]. Notably, thiourea/urea incorporating a nitrophenyl group as a signaling unit exhibits an enhanced hydrogen-bonding capability and proton release tendency [10]. The specificity of receptors incorporating one or more urea and thiourea subunits is dictated by the strength of the receptor–anion interaction. Robust hydrogen bond interactions form with anions harboring highly electronegative atoms, such as fluoride [11,12] or inorganic oxoanions [13,14]. Thiourea-containing polymers find diverse applications, including in heavy metal ion sorption [15,16], dye retention [17], optical chemical sensing for anion detection [18], and gene delivery [19].

Polyethyleneimine (PEI) emerges as a responsive synthetic polycation, boasting the highest concentration of amino groups in each macromolecular chain, with one amino group for every two methylene units. Branched PEI is commercially available in various molar masses and chain architectures (hyperbranched, branched, or linear) and typically presents a composition of approximately 1:2:1 primary–secondary–tertiary amine groups. These exceptional attributes render PEI intriguing for a plethora of potential applications, either in its pristine form [20] or as a constituent within composite materials (e.g., hydrogels [21,22], composite nanoparticles [23], and layer-by-layer thin films [24,25]). Emerging utilizations of PEI and PEI-derived materials span across diverse domains, including metal ion sensing [20,26], trinitrotoluene (TNT) [27], curcumin [28], and para-nitrophenol [29], and as a specific adsorbent matrix for treating heavy metals [30], nitrate ions, anionic dyes, phosphate anions, and organic pollutants in wastewater [31,32,33]. Additionally, PEI finds utility in drug delivery systems [24,34], antibacterial coatings [35,36], and catalysts [37]. Recently, Li et al. [19] introduced methyl-TU groups onto the PEI backbone, enabling interactions with DNA phosphate groups via hydrogen bonds. Ghiorghita et al. [38] investigated the flocculation of inorganic particles using a thiourea derivative of branched polyethyleneimine (PEITU), demonstrating the synergistic effect of hydrogen bonding and electrostatic attraction forces. 

Compared with small organic molecule probes that are less soluble in water, polymer ion probes are less demanding for the detection environment, and some can even detect ions in pure water. However, there are no reports on using (thio)urea polyethyleneimine as an anion recognition tool. Here, we designed and synthesized polyethyleneimine-based colorimetric anion receptors for three main reasons: (i) Polyethyleneimine contains many free and active groups of amino (–NH_2_). Multiple thiourea and amino groups are good hydrogen bond donors for the construction of polyethyleneimine-based anion receptors. It is known that stronger interactions in the host–guest systems are usually a consequence of a larger number of H-bonds in the complex [9]. Therefore, the complex stability highly depends on the strength and the number of the H-bonds. (ii) Polyethylenimide copolymer with low charge density can be easily reacted with isothiocyanate. (iii) The 4-nitrophenyl group is appended to the thiourea moiety, and the presence of –NO_2_^−^ electron-withdrawing group enhances the acidity and colorimetric recognition ability of the receptor.

In this paper, a nitrophenyl thiourea–polyethyleneimine (NTU−PEI) receptor L was synthesized via the reaction of PEI with 4-nitrophenyl thioisocyanate. The receptor’s structure was characterized using infrared spectroscopy and nuclear magnetic resonance hydrogen spectroscopy, while its anion-sensing properties were investigated via UV/vis spectroscopy. Notably, the receptor’s remarkable color response enabled the straightforward determination of its sensing behavior with the naked eye. TU derivatives were easily synthesized using a straightforward reaction involving an amino group and isothiocyanate [39]. In this study, NTU−PEI was prepared via an addition reaction between the amino group in PEI and the reactive isocyanate group of NPTI, as illustrated in Figure 1.

## 2. Materials and Methods

### 2.1. Materials

Branched polyethyleneimine (PEI) (average Mw: ~800 with LS and ~600 with GPC) was purchased from Shanghai Macklin Biochemical Technology Co., Ltd. (Shanghai China) 4-nitrophenyl isothiocyanate (NPTI) and the tetrabutylammonium (TBA) salts of all anions (with [(Bu)_4_N]_2_SO_4_] in a 50% aqueous solution) were supplied by Alfa Aesar and employed as received. Dichloromethane anhydrous was purchased from Hengyue Chemical Technology Co., Ltd. (Shanghai, China). All other reagents utilized in the experiment were of analytical grade and used without additional purification. In brief, 3 mmol of PEI (based on the structural unit’s molar mass) was introduced into a two-necked flask that had been evacuated under vacuum and purged with nitrogen 3 times. The PEI was dissolved in 30 mL of dry dichloromethane via magnetic stirring. Subsequently, 6.0 mmol of NPTI was dissolved in 20 mL of dry dichloromethane and added dropwise to the reaction mixture, followed by stirring at ambient temperature for 48 h. Afterward, stirring was ceased, and the solvent was evaporated under reduced pressure, resulting in 5 mL of the residual solvent. The reaction mixture was dialyzed against distilled water for five days. Lastly, the purified polymer was freeze-dried using a Fu Rui LGJ-10C device.

### 2.2. Methods

#### 2.2.1. Fourier-Transform Infrared (FT-IR) Spectroscopy

The solid-state FT-IR spectral data of PEI and NTU−PEI were obtained using a BRUKER EQUINOX-55 spectrometer (Ettlingen, Germany). The spectral data were acquired in the 4000–400 cm^−1^ range utilizing KBr pellets. The 5 mg samples were then filtered using a Millex-HV PVDF 0.45 µm, 33 mm Before analysis, all specimens were vacuum-dried at 60 °C for 24 h.

#### 2.2.2. Nuclear Magnetic Resonance (NMR) Spectroscopy

Liquid-state ^1^H NMR spectroscopy was employed to elucidate the chemical structures of PEI and NTU−PEI. The ^1^H NMR spectral data were acquired utilizing a VARIAN INOVA-400 spectrometer (400 MHz, Palo Alto, CA, USA), with a 5 mg sample dissolved in 0.6 mL DMSO-d_6_.

#### 2.2.3. UV/vis Absorption Titration

Absorption spectral data were acquired using a UV-6100S UV/vis spectrophotometer (Shanghai, China). The receptor solution concentration was determined based on the monomeric units of NTU−PEI (4.0 × 10^−5^ M) in DMSO/H_2_O (9:1 volume ratio), and the guest anions (2.0 × 10^−3^ M) were prepared in DMSO. For titration, small aliquots (2–5 μL) of the guest ion titration specimens were added to the receptor solution (2 mL), and the spectral data were scanned after each addition. All measurements were conducted at ambient temperature.

#### 2.2.4. Absorption Titrations and Hill Plots [40]

Sample solutions containing different levels of anions were prepared following a similar procedure, and their UV/vis absorption values were determined to monitor changes. A titration curve was generated by plotting the absorbance at the appropriate wavelength against the ratio of [anion] to [monomeric units of receptor]. The binding data were subsequently determined using the Hill Equation (1):(1)log[Y(1−Y)]=nlog[anion]+nlogKa
where *Y*, *n*, and *K*_a_ denote the fractional saturation of the host, the Hill coefficient (the average number of anions bound by each receptor molecule), and the apparent binding constant, respectively. The *Y*-value was determined using Equation (2):(2)$$Y=\frac{\Delta A_{obs}}{\Delta A_{\max}}=\frac{\left(A_{obs}-A_{0}\right)}{\left(A_{\max}-A_{0}\right)}$$
where *A*_0_, *A*_max_, and *A_obs_* denote the inherent absorbance of the urea–chitosan sample, the maximum absorbance reached upon the completion of absorption changes at the selected wavelength, and the absorbance in the presence of the guest anion, respectively. The slope *n* and *K*_a_ values were calculated based on the slope and *Y*-intercept in the obtained Hill plots.

#### 2.2.5. Solubility Test

The solubility of the compound nitrothiourea polyethyleneimine was tested in common organic solvents such as dimethyl sulfoxide (DMSO), dimethylformamide, distilled water, dichloromethane, ethanol, and acetonitrile at 25 °C. The samples were immersed in each solvent at a concentration of 10 mg/mL.

#### 2.2.6. Test Paper Experiment

Filter paper was cut into 4 × 4 squares, immersed in the receptor solution (1 × 10^−3^ mol/L) for 6 h, and then air-dried. Subsequently, the filter paper was labeled with “F^−^” using fluoride ion solutions of 0.1 mol/L, 0.5 mol/L, and 1 mol/L, respectively. The color changes on the filter paper were observed, and a visual recognition experiment for the anion was conducted.

## 3. Results and Discussion

### 3.1. FT-IR Analysis

Figure 1 illustrates the FT-IR spectral data of PEI and NTU–PEI. In the spectrum of PEI, characteristic peaks are observed at (i) 3282 and 3348 cm^−1^ for symmetric and asymmetric stretching vibrations of the –N–H group, (ii) 2823 and 2939 cm^−1^ for symmetric and asymmetric stretching vibrations of the –C–H group, (iii) 1666 cm^−1^ for bending vibrations of the –NH_2_ group, (iv) 1461 and 1593 cm−1 for bending vibrations of the –CH_2_ and –N–H groups, (v) 1299 and 1353 cm^−1^ for C–H bending vibrations, and (vi) 1049 and 1122 cm^−1^ for –C–N– and –C–C– stretching vibrations, respectively [41]. In contrast, the spectrum of NTU-PEI exhibits slight changes owing to incorporating NTU moieties. The peak centered at 1569 cm^−1^ may be ascribed to a combination of the bending vibrations of –NH_2_ groups and the stretching vibrations of –NH–C=S groups (previously reported at 1539 cm^−1^) [19,38]. The distinct peak at 1176 cm^−1^ corresponds to the stretching vibrations of –C=S groups [42]. Other notable peaks include those at 1326 cm^−1^ (–NO_2_), 1504, 852, 752, 717, and 655 cm^−1^ (phenyl) [18]. Furthermore, several peaks exhibit slight downshifts, indicating altered electronic effects from integrating NTU groups into the PEI backbone.

### 3.2. ^1^H NMR Analysis

In Figure 2a, the ^1^H NMR spectrum of PEI displays a signal ranging from 2.50 to 3.00 ppm, representing the characteristic broad resonance associated with branched groups, including methylene groups (N–CH_2_, CH_2_–NH–CH_2_, and CH_2_–NH_2_). Proton peaks corresponding to amine groups are evident from 1.10 to 2.10 ppm, encompassing –NH_2_ and CH_2_–NH–CH_2_ [18,43]. Additionally, solvent hydrogens exhibit a singular peak at approximately 7.27 ppm.

In Figure 2b, the ^1^H NMR spectrum of NTU-PEI reveals aromatic proton peaks ranging from 7.50 to 8.50 ppm, characteristic of mono-substituted benzene derivatives such as δ 1.90–1.40 ppm (–CH_2_) and δ 3.45 ppm (–NH), respectively. Notably, a new absorption peak at 2.64 ppm signifies the chemical shift of the amine proton of the introduced thiourea group to the PEI amine group. Similar observations have been documented in the literature [44,45]. The combined findings of the FT-IR and ^1^H NMR analyses affirm the successful synthesis of NTU−PEI with varying NTU contents.

### 3.3. UV/vis Spectrophotometric Titration

The receptor’s anion-sensing capability was evaluated using UV/vis spectrophotometric approaches in a DMSO/H_2_O solution. Absorption titration was conducted with the receptor (4.0 × 10^−5^ M) in DMSO/H_2_O, where 2.0 × 10^−3^ M of tetrabutylammonium sulfate, fluoride, and acetate were incrementally added to the DMSO, up to 10 equiv. The resulting spectra for all tested anions are depicted in Figure 3. A prominent absorption peak at 394 nm was observed in the UV/vis spectral data of the receptor in DMSO/H_2_O. The experimental findings indicate that the receptor L exhibited either minimal or no response to tetrabutylammonium salts such as CI^−^, Br^−^, I^−^, NO_3_^−^, PF_6_^−^, and H_2_PO_4_^−^. However, notable color changes were observed after adding 10 equivalents of SO_4_^2−^, F^-^, and AcO^−^ under the same conditions.

Significant alterations in the UV/vis absorption spectral data were detected by introducing SO_4_^2−^ into the receptor solution, indicating the complexity between the receptor and the anion. The intramolecular charge transfer (ICT) band at 394 nm for the receptor is depicted in Figure 4a. This band gradually diminished as complexation progressed, with the emergence of a new peak at 520 nm (bathochromic shift). Isosbestic points were discerned at 326 nm and 428 nm for L. The saturation of the absorbance value of L was achieved by adding 27 equivalents of SO_4_^2−^.The detection limit was 4.68 × 10^−7^ M, as determined by colorimetric data [46].

Figure 5a demonstrates that the absorption peak at 394 nm gradually diminished while a new peak emerged at 518 nm with increasing concentrations of F^−^ anions. Isosbestic points were detected at 332 nm and 454 nm. The absorbance value of L reached saturation by adding 30 equivalents of F^−^. The detection limit was calculated to be 5.12 × 10^−7^ M for the F^−^.

Figure 6a demonstrates that increasing concentrations of AcO^−^ anions led to a gradual decline in the absorption peak at 398 nm, along with the appearance of a new peak at 524 nm. Isosbestic points were discerned at 290 and 410 nm. The absorbance value of L reached saturation with 7.6 equivalents of AcO^−^. AcO^−^ with a detection limit of 3.26 × 10^−6^ M. Incorporating other anions as their TBA salts elicited no significant response, underscoring the receptor’s strong affinity for preferential binding interactions with SO_4_^2−^, F^−^, and AcO^−^ anions. These findings affirm the receptor’s higher selectivity for fluoride and sulfate over other anionic species.

### 3.4. Recognition Performance

The visual detection experiments were first conducted in DMSO/H_2_O by introducing the corresponding anions (2.0 × 10^−3^ M) into solutions of the receptor (1.0 × 10^−5^ M). Discernible color alterations from colorless to yellow occurred upon adding TBA fluoride anions into the DMSO solutions of the receptor. This change was accompanied by the emergence of broad new bands centered at 516 and 512 nm in the receptor’s UV/vis spectral data, indicating a stronger interaction between fluoride and the receptor owing to its smaller size and higher electronegativity than other halides. The selective color change induced in the receptor with dihydrogen sulfate and acetate resulted in a distinct yellowish hue upon adding SO_4_^2−^and AcO^−^ (1.0 × 10^−5^ M) (Figure 7). The observed alterations in color are likely attributed to the establishment of hydrogen bond interactions between the thiourea groups and the respective anions [47].

A colorimetric detection assay was carried out in DMSO/H_2_O upon adding various anions to the receptor solution (1 × 10^−5^ M). Incorporating 10 equivalents of SO_4_^2−^, F^−^, and AcO^−^ resulted in observable color changes—yellow for F^−^ and yellowish for SO_4_^2−^ and AcO^−^ (Figure 7)—which were discernible with the naked eye. However, no color changes were observed with CI^−^, Br^−^, I^−^, NO_3_^−^, PF_6_^−^, and H_2_PO_4_^−^. This lack of response may be due to the strong hydrogen bonding between SO_4_^2−^, F^−^, and AcO^−^ with a high negative charge density and the -NH group in the receptor.

### 3.5. Hill Plots

The binding constants (*K*_a_) of L for the SO_4_^2−^, F^−^, and AcO^−^ anions were determined in DMSO via UV/vis titration experiments based on 1:1 stoichiometry. The findings are summarized in Table 1. Hill plots illustrating SO_4_^2−^, F^−^, and AcO^−^ are presented in Figure 4b, Figure 5b and Figure 6b. The following binding constants were obtained from the slopes and intercepts of the Hill plots: *n* = 1.59 and *K*_a_ = 7.19 × 10^3^ M^−1^ for L/SO_4_^2−^ (R^2^ = 0.992), *n* = 1.7 and *_K_*_a_ = 6.87 × 10^3^ M^−1^ for L/F^−^ (R^2^ = 0.993), and *n* = 1.08 and *K*_a_ = 2.38 × 10^3^ M^−1^ for L/AcO^−^ (R^2^ = 0.983). These results demonstrate L’s strong binding affinity to form stable complexes with SO_4_^2−^, F^−^, and AcO^−^. An *n*-value exceeding one indicates positive cooperativity between L and SO_4_^2−^, F^−^, and AcO^−^ [48].

### 3.6. Solubility of NTU−PEI

The solubility of PEI and NSPEI in different solvents was studied at 25 °C (Table 2). As shown in the table, PEI is only insoluble in DMSO and can be dissolved in other organic solvents and water. NTU−PEI is insoluble in ethanol and acetonitrile and shows good solubility properties in both polar nonprotonic solvents and protonic solvents.

### 3.7. Test Paper Experiment Analysis

As the concentration of anions increases, the color of the font gradually intensifies. Therefore, while qualitatively detecting anions, it is also possible to preliminarily quantify the magnitude of the anion concentration based on the depth of the color, thereby enhancing the versatility of Receptor L in practical applications (Figure 8). This gradual intensification of color is attributed to the increased formation of receptor–anion complexes.

## 4. Recognition Mechanism

When acceptor molecules coexist with anionic F^−^, the solution turns yellow. With the coexistence of acceptor molecule L and the anion F^−^ in the solution, introducing a small quantity of methanol intensified the yellow coloration with the increasing methanol quantity. Simultaneously, the absorption peak at 394 nm gradually increased until reaching a state without anions. Concomitantly, the absorption peak at 518 nm gradually diminished. This phenomenon arose from the competition between methanol molecules and the anionic acceptor molecules for hydrogen bonding sites, highlighting the hydrogen bonding interactions between the guest and host.

The nitrophenyl thiourea-modified polyethyleneimine receptor demonstrated selectivity toward SO_4_^2−^, forming stable complexes. This may be because (i) incorporating the potent electron-withdrawing group (–NO_2_) onto the aromatic moiety within receptor L enhanced the thiourea NH proton’s ability and increased the SO_4_^2−^ complex’s stability; (ii) numerous thiourea and amino groups present in the receptor allowed multiple hydrogen bond donors to form, with the complex’s stability contingent on the number and strength of hydrogen bonds; (iii) the basicity of the sulfate anions significantly contributed to the selectivity of NTU-PEI for sulfate anions; and (iv) thiourea can act as an oxoanion receptor due to its ability to establish two N-H O bonds with the adjacent oxygen atoms of the anion. Figure 9 illustrates the potential complexes formed with the anion.

Fluoride is unique as it is similar to oxygen in size but carries a net negative charge. Among anions, fluoride forms the most robust hydrogen bond interactions with the NH fragment of the thiourea subunit, potentially involving an advanced phase of proton transference. These results can be attributed to the elevated charge density, diminutive size, and increased basicity. Moreover, the dual-deprotonation of the thiourea receptor by fluoride is conceivable due to the robust electron-withdrawing groups on the receptor.

## 5. Conclusions

In summary, a simple colorimetric receptor was successfully synthesized, and here, a nitrophenyl group was treated as a signaling unit and thiourea group as the binding sites. The anion recognition properties via hydrogen-binding interactions were investigated by UV–vis titrations, and the spectrum could be easily changed on addition of anions. The sensor exhibited responsiveness upon introducing SO_4_^2−^, F^−^, and AcO^−^ anions in DMSO/H_2_O (9:1) solution, leading to visible color alterations with the naked eye:from colorless to yellow for F^−^, and yellowish for SO_4_^2−^, and AcO^−^ at ambient temperature. No response was observed with the other anions such as CI^−^, Br^−^, I^−^, NO_3_^−^, PF_6_^−^, and H_2_PO_4_^−^. The underlying mechanisms of this receptor may involve hydrogen bonding interactions.

## Data Availability

All data generated or analyzed during this study are included in this published article.

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
