# Peer review of "Nitrophenyl Thiourea-Modified Polyethylenimine Colorimetric Sensor for Sulfate, Fluorine, and Acetate"

_sensors, 2024, doi:10.3390/s24123751_

Round 1

Reviewer 1 Report

Comments and Suggestions for Authors

Reviewer Feedback to the Author:

The experimental methodology needs strengthening to ensure greater rigor and scientific robustness, crucial for effectively engaging readers. While the manuscript explores the detection of various ions using a Nitrophenyl Thiourea-Modified Polyethylenimine-based fluorescent probe, it lacks sufficient discussion on practical applications and detailed insights into sensing mechanisms.

Author Response

Thank you very much for your comments and professional advice. These opinions help to improve academic rigor of our articale.

According to reviewers’ comments we made revision to original manuscript. We have studied comments carefully and have made correction which we hope meet with approval. Revised portions are marked in red in the paper. The main corrections in the paper and the responds to the reviewer’s comments are as following:  "Please see the attachment."

  1. The title of the study, "Nitrophenyl Thiourea-Modified Polyethylenimine: Synthesis and Anion Recognition," provides information but could be refined for clarity.

Response: For clarity, the title of this study has been revised to read "Nitrophenyl Thiourea-Modified Polyethylenimine -based  colorimetric Sensor for Sulfate ,fluorine and acetate" .

  1. The experimental methodology needs strengthening to ensure greater rigor and scientific robustness, crucial for effectively engaging readers. While the manuscript explores the detection of various ions using a Nitrophenyl Thiourea-Modified Polyethylenimine-based fluorescent probe, it lacks sufficient discussion on practical applications and detailed insights into sensing mechanisms.

Response: We have added Test paper experiment. These additions were marked in red in revised Manuscript.

2.2.6. Test paper experiment

Filter paper was cut into 4×4 squares, immersed in the receptor solution (1×10-3 mol/L) for 6 hours, and then air-dried. Subsequently, the filter paper was labeled with "F-" using fluoride ion solutions of 0.1 mol/L, 0.5 mol/L, and 1 mol/L, respectively. The color changes on the filter paper were observed, and a visual recognition experiment for the anion was conducted.

3.7. Test paper experiment analysis

As the concentration of anions increases, the color of the font gradually intensifies. Therefore, while qualitatively detecting anions, it is also possible to preliminarily quantify the magnitude of the anion concentration based on the depth of the color, thereby enhancing the versatility of Receptor L in practical applications. This gradual intensification of color is attributed to the increased formation of receptor-anion complexes.

      Fig. 8.  Photographs of different concentration of  F–on receptor R test paper 

  1. One limitation of the proposed probe is its development in solvents like DMF andDMSO, which dissolve the ions of interest. It's essential to address how this approach achieves enhanced sensitivity and selectivity, especially considering the typical dispersion or dissolution of these ions in an aqueous medium for real-world applications. Detailed discussions on these aspects are warranted.

Response: We have added organo-solubility of the modified chitosan in Table 2. These additions were marked in red in revised Manuscript.

2.2.5. Solubility Test

The solubility of the compound nitrothiourea polyethyleneimine was tested in common organic solvents such as dimethyl sulfoxide (DMSO), dimethylformamide, distilled water, dichloromethane, ethanol, and acetonitrile at 25°C. The samples were immersed in each solvent at a concentration of 10 mg/mL.

3.6. Solubility of NTU-PEI

The solubility of PEI and NSPEI in different solvents was studied at 25°C (Table 2). As shown in the table, PEI is only insoluble in DMSO and can be dissolved in other organic solvents and water. NTU-PEI is insoluble in ethanol and acetonitrile, and shows good solubility properties in both polar nonprotonic solvents protonic solvents.

Table 2. Solubility properties of NTU-PEI

Samples

Solubilitya

DMSO        DMF        H2O        CH2Cl2            C2H5OH                   CH₃CN

PEI

NTU-PEI

-           +                   +

+            +                   +

+                        +        

±                       -

+

- 

a (+: soluble; ±: partially soluble; -: insoluble)

  1. The introduction section should offer clearer insights into the limitations of previous studies and articulate how the current research overcomes these shortcomings in terms of sensitivity and selectivity. Expanding on the advantages of this study would provide a more comprehensive understanding.

Response: We have added current research overcomes these shortcomings in terms of sensitivity and selectivity.  These additions were marked in red in revised Manuscript.

Compared with small organic molecule probes that are less soluble in water, polymer ion probes are less demanding for the detection environment, and some can even detect ions in pure water However, there are no reports on using (thio)urea polyethyleneimine as an anion recognition tool. Here we designed and synthesized polyethyleneimine based colorimetric anion receptors for three main reasons: (i) polyethyleneimine contains many free and active groups of amino (–NH2) . Multiple thiourea, and amino groups are good hydrogen bond donors for the construction of the polyethyleneimine based anion receptors. It is known that stronger interactions in the host-guest systems are usually a consequence of a larger number of H-bonds in the complex[39].Therefore, the complex stability highly depends on the strength and the number of the H-bonds. (ii) poly(vinylamine) is a macromolecule with low site resistance, which can be easily reacted with isothiocyanate and (iii) the 4-nitrophenyl group was appended to the thiourea moiety, and the presence of –NO2– electron-withdrawing group enhanced the acidity, and colorimetric recognition ability of the receptor .

  1. Delving deeper into the functional groups involved in fluorescence quenching by the Nitrophenyl Thiourea-Modified Polyethylenimine-based fluorescent probe,along with providing a detailed mechanism of quenching and desorption regeneration, would enhance the manuscript's quality.

Response: Thank you very much for your comments and professional advice. We believe that if the structure of a nitrophenyl thiourea modified polyethylenimine probe is analyzed, the probe does not contain fluorophores, and the fluorescence performance is very weak or does not have fluorescent properties, so it is not necessary or will continue to study the functional groups of fluorescence quenching, as well as the quenching and desorption regeneration mechanisms in future work.

  1. Figure 1 should include discussions on FT-IR spectra, highlighting each spectral study with the proposed ions before and after fluorescent quenching using the novel probe.Furthermore, discussions on how these ions are affected during the studies should be incorporated.

Response: Thank you very much for your comments and professional advice. We believe that nitrophenyl thiourea modified polyethylenimine probe structure does not contain fluorophores and has very weak or no fluorescent properties, so there is no discussion in Figure 1 for each of the spectral studies of the ions presented before and after fluorescence quenching of the novel probe in FT-IR spectroscopy. In addition, how these ions are affected during the study should also be discussed" for further study in subsequent work.

  1. I recommend including a schematic representation illustrating the mechanism of interaction between the ions and the fluorescent quenching by the probe. Please discuss the possible functional groups involved in these interactions.

The following research article may provide further understanding: "An Onepot synthesis of carbon dots from neem resin and their selective detection of Fe(II) ions and photocatalytic degradation of toxic dyes, RSC Sustainability 2024, 2, 635-645.".

Response: The probe does not contain a fluorophore, and I don't think it is necessary to study or further investigate the "interaction mechanism between ions and fluorescence quenching of probes" in subsequent work.

  1. Please provide the Limit of Detection (LOD) and Limit of Quantification (LOQ) for each ion proposed by this study.

Response: The detection limit was 4.68×10-7 M for SO42 , as determined by colorimetric data [47]. The detection limit was calculated to be 5.12 ×10-7 M for the F. AcO with a detection limit of 3.26×10-6 M.

  1. Regeneration studies of the fluorescent probes should be conducted using appropriate desorbing solutions, and the results should be showcased with digital images before and after regeneration. Additionally, FT-IR characterization studies should be provided.

Response: The stability of the probe are still being investigated.

  1. To facilitate future research, consider addressing detailed mechanistic studies, optimizing fluorescent probes, conducting long-term stability assessments, analyzing scale-up and cost implications, and evaluating environmental impacts. Addressing these aspects can significantly contribute to the development of efficient and sustainable fluorescent probes for water purification.

Response:  The more detailed study of the optimizing fluorescent probes, conducting long-term stability assessments, analyzing scale-up and cost implications, and evaluating environmental impacts are still being investigated.

Reviewer 2 Report

Comments and Suggestions for Authors

This manuscript presents the synthesis of a NTU-PEI receptor as colorimetric sensing element for thiourea. The chemical synthesis, characterization via spectroscopic methods and application (proof-of-concept) seem sound.

Specific comments:

- Please indicate the sample size (i.e. n) for the characterization of the sample of the receptors in the Methods section.

- Conclusions section can be improved as it is too summarized.

Author Response

Thank you very much for your comments and professional advice. These opinions help to improve academic rigor of our articale. Based your suggestion and request , we have made corrected modifications an the revised manusccript. "Please see the attachment." 

  1. Please indicate the sample size (i.e. n) for the characterization of the sample of the receptors in the Methods section.

Response: First,Fourier-transform infrared (FT-IR) spectroscopy in the Methods section,We have added. These additions were also marked in red in revised Manuscript.  Second, the 2.2.2. Nuclear magnetic resonance (NMR) spectroscopy in the Methods section,We have added. These additions were also marked in red in revised Manuscript.

  1. Conclusions section can be improved as it is too summarized.

Response:  The conclusion section has been improved,  These additions were also marked in red in revised Manuscript.

In summary, a simple colorimetric receptor was successfully synthesized and here nitrophenyl was treated as a signaling unit and thiourea group as the binding sites.The anion recognition properties via hydrogen-binding interactions were investigated by UV–vis titrations and the spectrum could be easily changed on addition of anions. The sensor exhibits responsiveness upon introducing SO42−, F, and AcO anions in  DMSO/H2O (9:1) solution, leading to visible color alterations with the naked eye: from colorless to yellow for F and yellowish for SO42− and AcO at ambient temperature. No response was observed with the other anions such as CI, Br, I, NO3, PF6, and H2PO4. The underlying mechanisms of this receptor may involve hydrogen bonding interactions.

If the revised manuscript needs further revision, please inform me any time.

Thank you in advance.

Yours sincerely,

Prof. Kuerbanjiang Rouzi
